# Boundary conductance in macroscopic bismuth crystals

Woun Kang [1,5], Felix Spathelf [2,3,5], Benoît Fauqué [3], Yuki Fuseya [4] & Kamran Behnia [2✉]

The interface between a solid and vacuum can become electronically distinct from the bulk. This feature, encountered in the case of quantum Hall effect, has a manifestation in insulators with topologically protected metallic surface states. Non-trivial Berry curvature of the Bloch waves or periodically driven perturbation are known to generate it. Here, by studying the angle-dependent magnetoresistance in prismatic bismuth crystals of different shapes, we detect a robust surface contribution to electric conductivity when the magnetic field is aligned parallel to a two-dimensional boundary between the three-dimensional crystal and vacuum. The effect is absent in antimony, which has an identical crystal symmetry, a similar Fermi surface structure and equally ballistic carriers, but an inverted band symmetry and a topological invariant of opposite sign. Our observation confirms that the boundary interrupting the cyclotron orbits remains metallic in bismuth, which is in agreement with what was predicted by Azbel decades ago. However, the absence of the effect in antimony indicates an intimate link between band symmetry and this boundary conductance.

[1] Department of Physics, Ewha Womans University, Seoul 03760, Korea. [2] LPEM (CNRS-Sorbonne University, ESPCI Paris, PSL University, Paris 75005, France. [3] JEIP (USR 3573 CNRS), Collège de France, PSL University, Paris 75231, France. [4] Department of Engineering Science, University of Electro-Communications, Chofu 182-8585 Tokyo, Japan. [5] These authors contributed equally: Woun Kang, Felix Spathelf. ✉email: kamran.behnia@espci.fr

Bismuth is a semimetal with an extremely low density of highly mobile carriers of both signs[1-3]. The long Fermi wavelength of its carriers extends over several tens of lattice parameters. Therefore, only extended defects (such as dislocations) can decay the charge current. In crystals lacking such spatially extended disorder, carriers become ballistic[4], and their mobility (and as we will see below their magnetoresistance) easily exceeds any other solid hitherto explored[5].

Recent research has shown that an extended Dirac Hamiltonian combined to the Fermi surface structure derived by a tight-binding model[6] can explain the complex angle-dependent Landau spectrum of electrons and holes[7]. The approach successfully accounts for the total evacuation of one or two electron pockets at strong magnetic fields aligned along different axes[8]. The angle-dependent magnetoresistance and its rich structure[9] are also accessible to semiclassical transport theory treating mobility as a tensor[9].

Open questions remain, however. The origin of the loss of rotational threefold symmetry in presence of strong magnetic fields[9-11] is yet to be understood. Such a 'nematicity' was also observed on the surface of bismuth crystals[12], as well as in other solids[13]. In addition to 'valley nematicity'[14,15], other theoretical possibilities for its origin were proposed[16].

Another open question is the topology of the electron wave function[17,18], its consequences for the metallic surface states in bismuth[19] and the evolution of the latter with thickness and Sb substitutions[20]. The topology of surface states and their status in the trivial/non-trivial dichotomy has been a subject of ongoing debate[21-27]. A recent popular theory identifies bismuth as a higher-order topological solid with topologically protected hinge states[23]. This hypothesis has been invoked to explain the experimental observation of ballistic transport in micrometer-long bismuth nanowires[28].

Here, we present a study of magnetoresistance in prismatic crystals of bismuth with ballistic carriers with unexpected consequences for both these issues. By choosing specific crystallographic planes as faces of the prisms, we uncover a specific contribution to electric conductivity when the magnetic field is aligned parallel to a two-dimensional boundary between the three-dimensional crystal and vacuum. The absence of this effect in antimony crystals of identical shapes points to the role played by the band structure topology in tuning the edge-bulk correspondence in macroscopic crystals in the high-field limit ($\omega_c \tau \gg 1$). It confirms that the interruption of cyclotron orbits at the boundary of a macroscopic three-dimensional crystal can generate a highly conducting boundary state in which bulk magnetoresistance is absent[29,30].

It is known that a periodically driven Hamiltonian can provide topological protection in the so-called Floquet systems[31-33]. In presence of quantizing magnetic fields, cyclotron orbits interrupted at the edge may be conceived as a periodical perturbation to the local electrons, but we are not aware of any available theory on this.

Our results identify the source of the loss of rotational symmetry and apparent 'nematicity'[9-11] in bismuth crystals. The existence of distinct edge states surrounding bulk states would also explain why identical bismuth tilted crystals across a twin boundary can keep different chemical potentials at high magnetic field[7].

($n = p = 3 \times 10^{17}$ cm$^{-3}$ in bismuth and $n = p = 5.5 \times 10^{19}$ cm$^{-3}$ in antimony[6]). For Bi, $\rho_0 = 0.18$ μΩ cm corresponds to an average mobility of $\langle \mu_e + \mu_h \rangle = 1.2 \times 10^8$ cm$^2$ V$^{-1}$ s$^{-1}$. For Sb, a residual resistivity of $\rho_0 = 0.07$ μΩ cm corresponds to an average mobility of $\langle \mu_e + \mu_h \rangle = 1.7 \times 10^7$ cm$^2$ V$^{-1}$ s$^{-1}$. These mobilities exceed those of the samples used in previous studies of magnetoresistance on Bi[8-10] and Sb[34].

Note that the average mobility deduced from residual resistivity ignores the fact that in presence of anisotropic Fermi pockets of electrons and holes, the electrons and holes in different pockets and along different orientations have different mobilities. It is safe to assume that some carriers are ballistic given the size dependence of the residual resistivity[35].

Carrier mobility in Bi is probably the highest known in any solid. Bi crystals with a RRR of $\simeq 600$ were reported in old scientific literature (Supplementary note 1, see Supplemental Material for more details). However, the high-field magnetoresistance of such samples was not reported before. Bismuth samples subject to pulsed magnetic fields[8,36] were small pieces cut from larger crystals and hosted extended scattering centers such as twin boundaries and dislocations. The presence of such disorder led to a shorter electronic mean free path and a significantly lower magnetoresistance compared to the crystals studied here.

The typical magnetoresistance in our crystals is shown in Fig. 1. One can see that the 14 T magnetoresistance of the Bi

**Table 1 Details of the samples used in this study.**

| Sample | Cross section | Face orientation | RRR | $\rho_0$ (nΩ cm) |
|---|---|---|---|---|
| Bi-Tri-1a | Triangle | $C_2$ | 323 | 400 |
| Bi-Tri-2a | Triangle | $C_1$ | 485 | 270 |
| Bi-Tri-1b | Triangle | $C_2$ | 576 | 220 |
| Bi-Tri-2b | Triangle | $C_1$ | 518 | 190 |
| Bi-Cub-1 | Square | Low symmetry | 393 | 260 |
| Bi-Cub-2 | Square | $C_1/C_2$ | 683 | 180 |
| Sb-Tri-1 | Triangle | $C_2$ | 3260 | 7.1 |
| Sb-Tri-2 | Triangle | $C_1$ | 3270 | 8.7 |

All cross sections were equilateral. Samples with the same type of cross section had identical dimensions (triangle: $4 \times 4 \times 4$ mm$^3$, square: (5 mm)$^3$). The face orientation refers to the crystallographic plane of the faces parallel to the orientation of the current which was always along the C3 axis, i.e., in Bi-Tri-1a, the C2 crystal axis is perpendicular to the three rectangular faces of the triangular prism. (See the insets of Fig. 4 for the visualization of the four types of geometry).

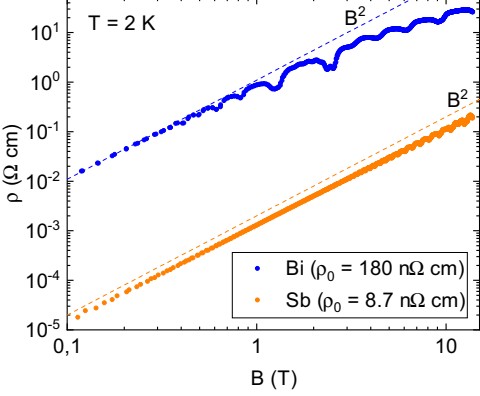

**Fig. 1 Amplitude of magnetoresistance.** The resistivity increases by 7 to 8 orders of magnitude upon applying a magnetic field of 14 T along a $C_1$ axis. In Bi, a downward deviation from $B^2$ behavior at high magnetic field is visible. For Bi-Cub-2, $\rho(B = 14$ T$)/\rho(B = 0) = 1.4 \times 10^8$ and for Sb-Tri-2, $\rho(B = 14$ T$)/\rho(B = 0) = 2.2 \times 10^7$.

## Results

### Samples, carrier mobility, and orbital magnetoresistance.
The bismuth (Bi) and the antimony (Sb) crystals used in this study are listed in Table 1. The residual resistivities are remarkably low, considering the low carrier density of these two semimetals

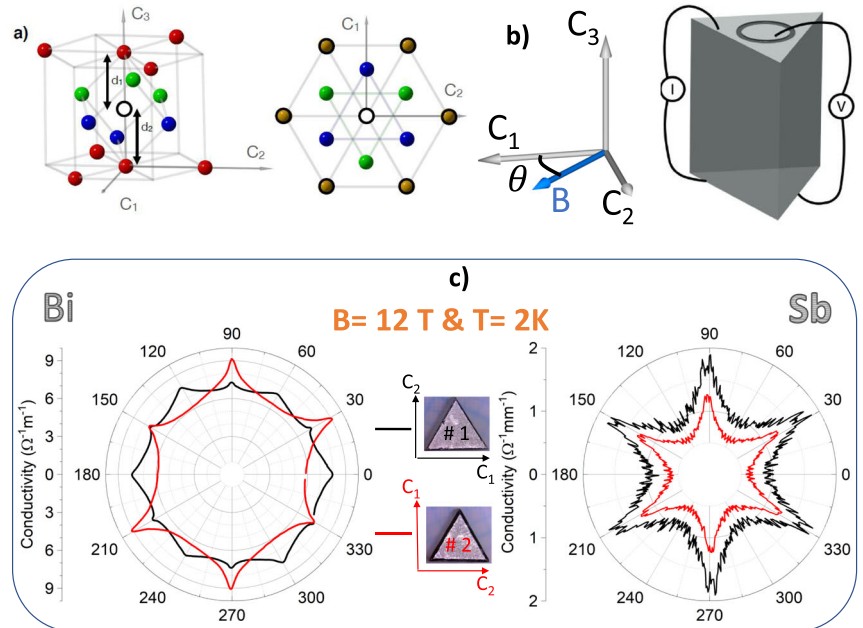

**Fig. 2 Crystal structure, triangular prisms, and angle-dependent magnetoconductance. a** Left: Rhombohedral crystal structure of bismuth and antimony. $C_1$, $C_2$, and $C_3$ refer to bisectrix, binary and trigonal axes. Note that all atoms are not shown and $d_1 > d_2$. Right: Projection to the trigonal plane. The central atom is surrounded by its first (in green), second (in blue), and third (in yellow) neighbors. Atoms with bold black rings belong to the same sub-lattice. **b** Experimental configuration for measuring angle-dependent magnetoresistance. The current electrodes were made large enough to cover most of the surface and the voltage electrodes were small circles. **c** Angle-dependent magnetoconductivity in two triangular prism-shaped crystals which are identical in shape, but whose faces are tilted by 30 degrees. In Bi (left), the low-temperature angle-dependent magnetoresistance is dissimilar in the two samples, but in Sb (right), they remain identical. Fine features in the angle-dependent conductivity of Sb are caused by evacuation of Landau levels upon rotation.

sample with RRR = 683 is orders of magnitude higher than what was observed in WTe$_2$ at 60 T[5,37]. The latter was dubbed 'extremely large magnetoresistance' by many authors[38]. However, such a large non-saturating magnetoresistance was reported by Kapitza back in 1928[39] and is a generic feature of low-density semimetals[40].

**Triangular prismatic crystals: Bi *vs.* Sb**. Bismuth and antimony crystallize in the rhombohedral A7 crystal structure shown in Fig. 2a. The three axes of high symmetry are known as trigonal (or $C_3$), binary ($C_2$), and bisectrix ($C_1$)[1,2]. As seen in Fig. 2b, in the trigonal plane, there are three $C_1$ and three $C_2$ axes, which are equivalent upon $2\pi/3$ rotation. Our experiments consisted in measuring the magnetoresistance of Bi and Sb crystals with the electric current applied along $C_3$ and the magnetic field rotating in the trigonal plane. The orientation of the magnetic field is given by $\theta$, which is defined as the angle between the field and the $C_1$ axis. As reported previously[9,10], despite the constant angle between current and field, orbital magnetoresistance varies with angle reflecting the in-plane anisotropy of the Fermi velocity in the three electron pockets.

Our main observation is illustrated in Fig. 2c. It shows the angular dependence of electrical conductivity $\sigma$ at a magnetic field of 12 T and 2 K in a pair of Bi crystals tailored identically as triangular prisms. (Note that since the Hall resistivity is negligible compared to the magnetoresistance, $\sigma \approx 1/\rho$). The only difference between the two crystals was that in one case each of the three square faces of the prism was a binary ($C_2$) plane, while in the other case, it was a bisectrix ($C_1$) plane. The angle-dependent magnetoresistance is clearly different in the two crystals. In one there are conductivity peaks each time the field is along a bisectrix axis. In the other, there are minima (instead of maxima) at the same field orientations.

The same experiment was performed in a pair of Sb crystals tailored in the same way as the bismuth crystals. As one can see in the figure, no difference is visible between the two crystals.

Thus, angle-dependent magnetoresistance depends on the choice of specific crystal planes as faces of each prism in bismuth crystals, but not in antimony crystals. We reproduced this observation in two other pairs of Bi crystals and one other pair of Sb crystals.

**Identifying the source of excess conductivity in bismuth**. The shape dependence of the orbital magnetoresistance sheds light on the origin of the loss of the threefold symmetry in bismuth crystals of various geometry[9,10]. As reported previously[9,10], this effect emerges only at sufficiently low temperature and high magnetic field. Let us now consider the amplitude of the required magnetic field.

Figure 3 shows the change of angle-dependent conductivity with temperature in a pair of Bi triangular prisms at two different fields, namely $B = 14$ T and $B = 0.2$ T. The non-trivial evolution of angle-dependent magnetoconductivity can be quantitatively described by invoking the anisotropy of the effective mass and the evolution of scattering time and carrier density among pockets with temperature and magnetic field. At $B = 0.2$ T and $T = 40$ K, angle-dependent magnetoconductivity displays a star-like shape. With cooling, the anisotropy is lowered due to a partial compensation of the mass anisotropy by the emerging anisotropy in the scattering time[4]. At $B = 14$ T, the anisotropy of mobility is compensated by an anisotropy in the distribution of carriers among the three pockets[8,9,41].

As can be seen in Fig. 3, at both fields, the two samples show an identical angle-dependent conductivity at 40 K, but not at low temperature. Upon cooling, additional features emerge in the first triangular prismatic crystal, which are absent in the second one. Now, at $B = 14$ T, all electrons are confined to their lowest

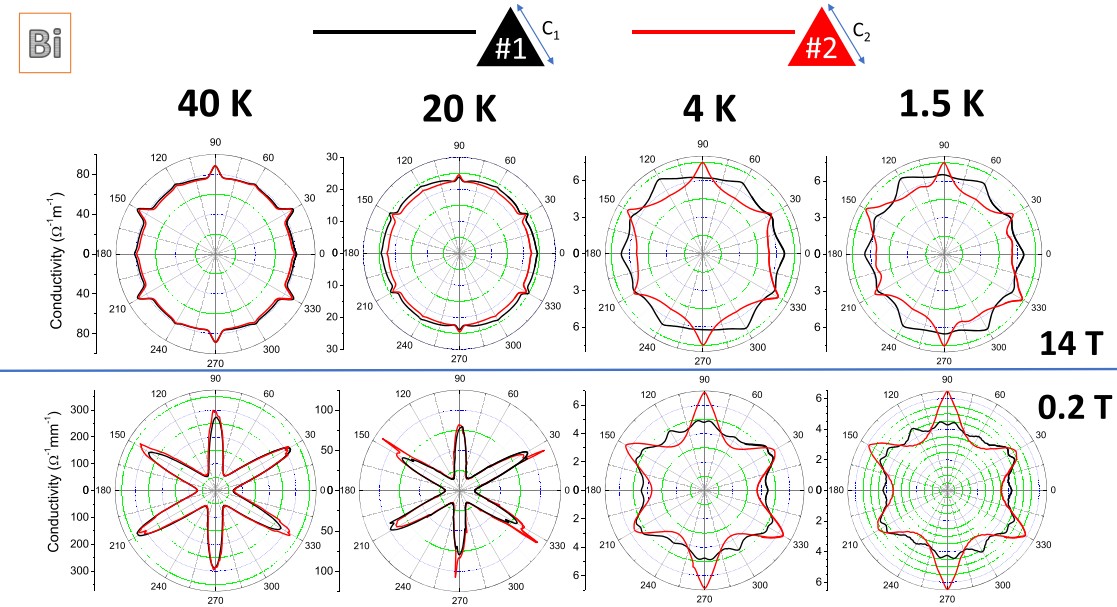

**Fig. 3 The emergence of shape dependence with cooling.** The evolution of angle-dependent magnetoresistance with cooling for $B = 14$ T (top panels) and for $B = 0.2$ T (bottom panels). Note the emergence of a difference at low temperatures in both cases.

Landau level, but not at $B = 0.2$ T. This implies that the observed shape dependence of magnetoconductivity does not require proximity to the quantum limit $\hbar\omega_c \approx E_F$, but the passage to the high-field limit $\omega_c\tau > 1$.

As for the high-field limit, two relaxation times are to be distinguished. The Dingle scattering time $\tau_D$, extracted from quantum oscillations is almost fifty times shorter than the transport scattering rate $\tau_{tr}$ in Bi[42]. Such a large difference between $\tau_{tr}$ and $\tau_D$ has been observed in several other dilute metals[35,43,44]. The semiclassical high-field limit ($\omega_c\tau_{tr} \approx 1$) is satisfied when the cyclotron radius becomes shorter than the mean free path. The quantum high-field limit ($\omega_c\tau_D \approx 1$) is satisfied when the distance between Landau levels becomes smaller than the broadening caused by temperature and disorder. At $T = 40$ K, the first criterion is satisfied, but not the second and there is no shape dependence. As the sample is cooled down, the shape dependence and quantum oscillations emerge concomitantly (Supplementary Note 2 and Supplementary Fig. 1, see Supplemental Material for more details). Therefore, one can safely conclude that what matters is the $\omega_c\tau_D \approx 1$ criterion.

The angle-dependent Landau spectrum in each sample is revealed by taking the second derivative of magnetoresistance. It remains identical in the two samples in spite of the difference in the sheer amplitude of the magnetoresistance (Supplementary Note 3 and Supplementary Fig. 2, see Supplemental Material for more details). This observation implies that their bulk Fermi surface is identical and can therefore be excluded as the origin of the shape dependence.

The origin of the additional features in the angle-dependent magnetoconductivity was pinned down by studying two other samples with a square cross section. Samples dubbed Bi-Cub-1 and Bi-Cub-2 (see Table 1) were cubic samples with identical dimensions. Both had two trigonal faces, but their four other faces were different. In Bi-Cub-2, the four other faces were two pairs of bisectrix and binary planes. On the other hand, in Bi-Cub-1, the pairs of faces other than trigonal were not aligned along a high-symmetry plane. They were rotated by a finite angle ($\approx\pi/4$) around the trigonal axis with respect to the two crystallographic planes (see insets in Fig. 4).

Figure 4 compares the angle-dependent magnetoconductivity of four Bi crystals with different shapes. The temperature and the magnetic field are identical in all cases and the current is always applied along the trigonal axis and the field is rotating in the trigonal plane. Two samples are prisms with triangular cross sections and two are cubes as detailed above. The magnitude of conductance is roughly similar in the four samples, which have comparable dimensions and mobilities. The striking difference is the presence of additional peaks in magnetoconductance and their angular locations. In all the samples, magnetoconductivity peaks when the magnetic field is along the binary axis ($B\|C_2$). On top of these peaks, in all the samples there is another set of peaks, which appear when the magnetic field is parallel to a face of the sample.

In the two triangular prisms, these peaks appear with a periodicity of $\pi/3$ and there are six of them. The difference between the two is that in one case the surface peaks and the binary peaks are concomitant and in the other there is a $\pi/6$ shift between the two sets of peaks. In prisms with a square cross section, on the other hand, the additional peaks appear with a periodicity of $\pi/2$ and there are four of them. They occur each time the field is parallel to one of the four faces. If this face happens to be parallel to the binary axis (i.e., if it is a crystallographic bisectrix plane), than the peak is more pronounced, as shown in Fig. 4d. One can also see that when distinct, the two types of peaks have different angular width and slightly different amplitudes. The surface peaks (marked by blue arrows) are typically twice wider and twice higher than the bulk binary peaks (marked by red arrows).

This observation clearly establishes that the additional contribution to conductivity emerges when the magnetic field (kept always perpendicular to the charge current flowing along the trigonal axis) lies parallel to a two-dimensional boundary of the three-dimensional sample. Moreover, it does not matter at all for this boundary to be a specific crystal plane. The magnitude of the additional contribution remains the same when the boundary in question is the binary plane, the bisectrix plane, or a low-symmetry plane.

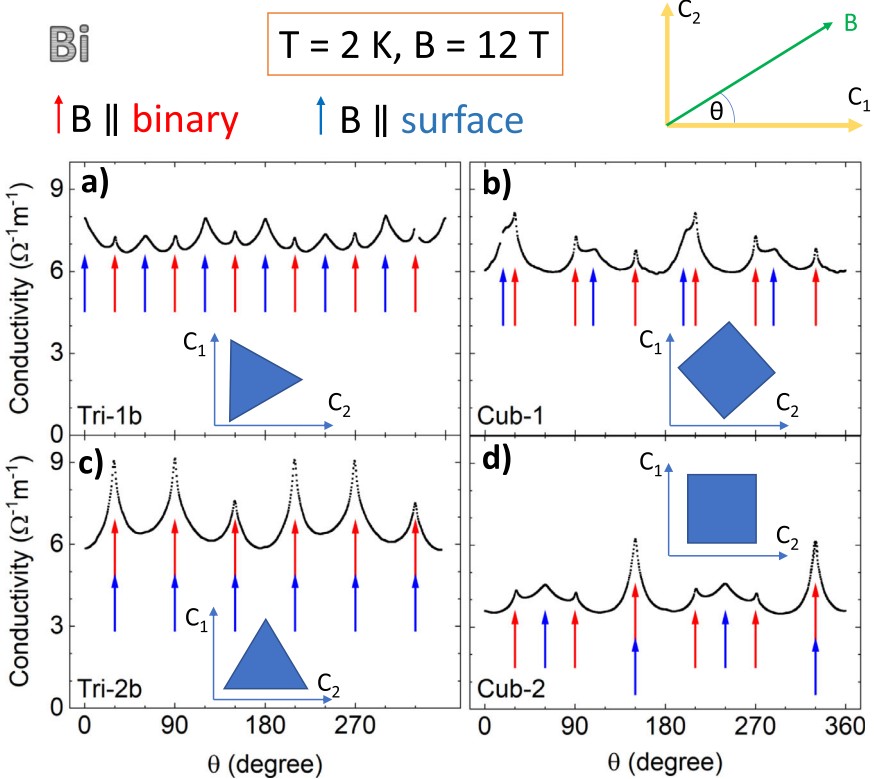

**Fig. 4 Angle-dependent conductivity peaks in samples with different shapes.** Angle-dependent conductivity at $T = 2\,K$ and $B = 12\,T$ in four different Bi samples **a** triangular prism Tri-1b; **b** triangular prism Tri-2b; **c** cubic sample Cub-1; **d** cubic sample Cub-2 (see Table 1). In all cases, conductivity peaks when the field is along a binary axis (red arrows) and when the field is parallel to a face of the polygon (blue arrows). This implies that the shape dependence is caused by an excess of conductance arising when the magnetic field is aligned with a flat boundary between the sample and vacuum.

**The boundary contribution and its relevant length scales.** Having demonstrated that the boundary conductivity emerges whenever the magnetic field is parallel to one of the surfaces of a prismatic crystal, let us now consider its evolution with magnetic field.

The relative contribution of the boundary conductance to the total conductivity can be estimated by subtracting the angle-dependent conductivity in two triangular prisms with different crystal planes as faces. This assumes that bulk magnetoconductivity is identical in the two, which is reasonable, but subject to caution given the slight difference in mobility, which implies a difference in the expected magnitude of orbital magnetoresistance.

Figure 5 a shows the relative change in the conductivity between sample Bi-Tri-1a and sample Bi-Tri-2a. What is shown is the evolution of $r = \frac{\sigma_{\#1} - \sigma_{\#2}}{\sigma_{\#1} + \sigma_{\#2}}$ with magnetic field and the angle between field and the crystal axes. The dimensionless $r$ alternates between 0.2 and −0.2. Vertical red stripes show the excess conductivity in sample 1 and vertical blue stripes show the excess conductivity in sample 2. Remarkably, the width of the stripes or their color does not vary with increasing magnetic field. The relative amplitude of the excess boundary conductivity does not change even when the field increases by two orders of magnitude and the amplitude of conductivity decreases by almost four orders of magnitude.

Figure 5b shows the dependence of $r$ on angle at $B = 0.2\,T$ and $B = 10\,T$. It is striking to observe how the two curves superpose on each other, in spite of a 50-fold change in magnetic field. Thus, the correction to conductivity brought by the B ∥ surface configuration both in amplitude and in angular dependence does

not evolve with magnetic length $\ell_B = \sqrt{\frac{\hbar}{eB}}$, which changes by a factor of 7 between the two fields.

The data presented in Fig. 5b contains another important feature. Within an angular window of ±4 degrees, we can fit each peak with a $\cos(q\theta)$ function with $q$ as a free fitting parameter. The fact that a simple cosine fits the data implies that the excess conductivity detected here is not singular, as observed in other contexts[45]. Moreover, we find that when the surface becoming parallel to the magnetic field was a bisectrix crystallographic plane $q = 20.3 \pm 3$ and when it was a binary crystallographic plane, it was $q = 8.1 \pm 1$. $q$ quantifies the sharpness of the peak, presumably caused by the anisotropy of the relevant length scales. Now, the Fermi momentum and wavelength of electrons is fourteen times longer along the bisectrix ($C_1$) axis than along the binary axis ($C_2$)[6]. This is a consequence of the huge (200-fold) anisotropy of the in-plane electron mass. Therefore, the significant difference between the angular width of the excess conductivity brings us to suspect a key role played by the Fermi wavelength of electron pockets in any plausible scenario.

## Discussion

**Cyclotron orbits and the 'static skin effect'.** Decades ago, Azbel and Peschanskii put forward the concept of a static skin effect at high magnetic fields in metals[29,30]. This idea provides the departing point of a plausible scenario to explain our observation. In the semiclassical picture, the magnetic field bends the electron trajectory. When the cyclotron radius is shorter than the mean free path of carriers, there is a large magnetoresistance. In semimetals, this magnetoresistance does not saturate even in the

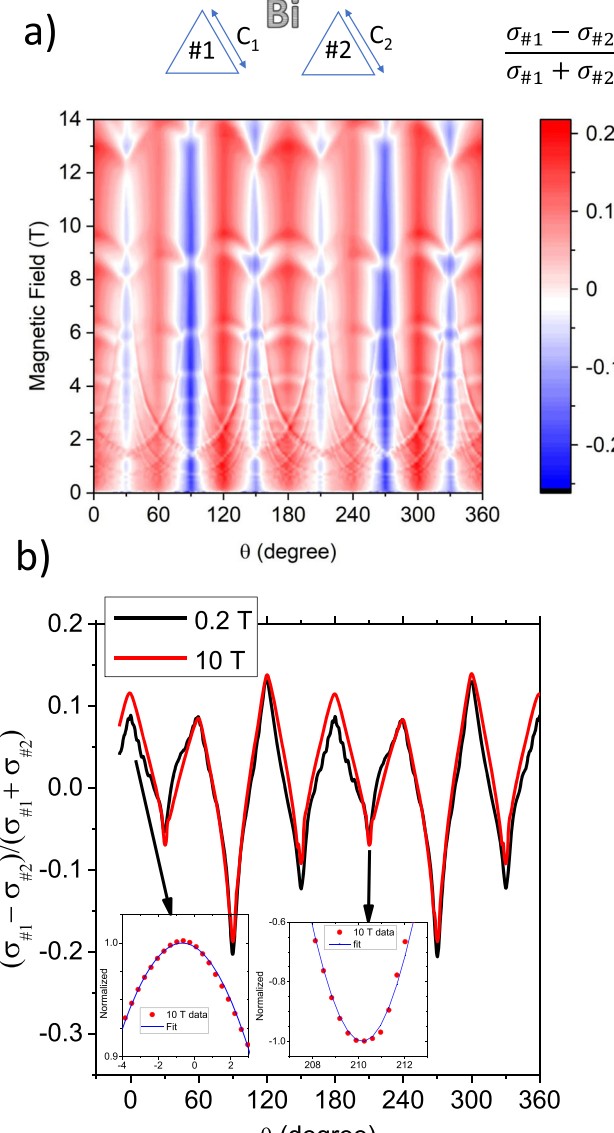

a)

$$\frac{\sigma_{\#1} - \sigma_{\#2}}{\sigma_{\#1} + \sigma_{\#2}}$$

b)

**Fig. 5 Field and angle dependence of the excess boundary conductance.**
**a** Color plot of $\frac{\sigma_{\#1}-\sigma_{\#2}}{\sigma_{\#1}+\sigma_{\#2}}$ at $T = 1.55$ K. Measurements were simultaneously
performed for both samples in a fixed magnetic field at intervals of 0.1 T
between 0 and 14 T. Values of $\frac{\sigma_{\#1}-\sigma_{\#2}}{\sigma_{\#1}+\sigma_{\#2}}$ were calculated at each pair of
angle and field values and put into a matrix of 721 × 141 dimensions. A
commercial software (Origin from OriginLab Corp.) was used to generate
the color contour map. Red and blue stripes represent the excess and the
deficit of conductivity. **b** Angle dependence of $\frac{\sigma_{\#1}-\sigma_{\#2}}{\sigma_{\#1}+\sigma_{\#2}}$ at $B = 0.2$ T and at
$B = 10$ T. The relative deficit and excess conductivity caused by field-
boundary alignment does not change significantly in spite of three orders of
magnitude change in the amplitude of bulk magnetoconductance. The angle
dependence also remains roughly identical for $B = 0.2$ T and $B = 10$ T.
Insets in panel b show $\cos(q\theta)$ fits to the data over a narrow angular
window (see text).

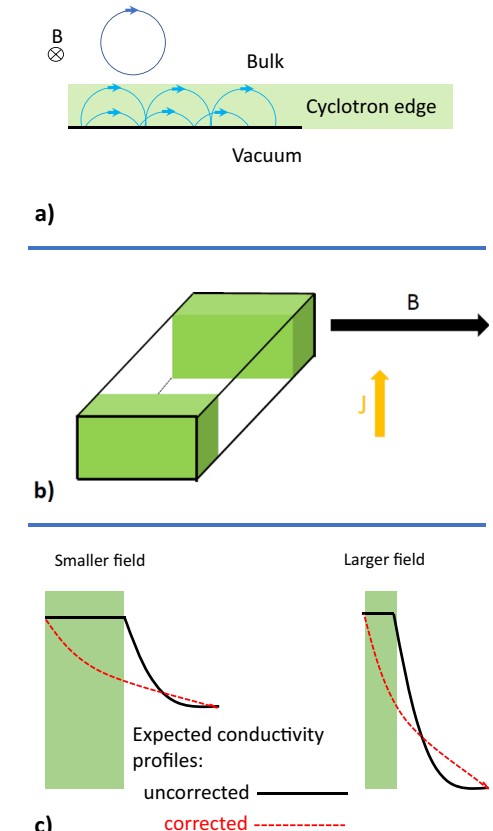

**Fig. 6 Static skin effect. a** When $\omega_c\tau \gg 1$, bulk carriers whirl along cyclotron
orbits numerous times without being scattered. This yields a semiclassical
account of the large orbital magnetoresistance in compensated semimetals
with ballistic carriers like Bi. The cyclotron orbits are interrupted at the edge
of the sample (in green). When reflections are specular, magnetoresistance
is canceled in this region. **b** In this semiclassical picture, the excess of
conductance when the field is parallel to a surface arises thanks to
additional conduction along dissipation-free edges. **c** In a larger magnetic
field, the cyclotron edge is narrower and the difference between bulk and
boundary conductivities is larger. Therefore, the conductivity profile is
expected to evolve with increasing magnetic field. It is sketched for two
different possibilities: (i) the conductivity inside the cyclotron edge does
not evolve with depth (solid black line), and (ii) hybridization leads to a
smooth variation across the cyclotron edge (red dashed line).
Experimentally, the relative conductivity excess does not change
above 0.2 T.

high-field limit. Now, at the surface of the sample, the cyclotron
orbits are interrupted and what matters is the scattering of the
carriers by the edge. If their collision results in a specular
reflection then the conductivity at the edge is much higher. As a
result, most of the current will flow near the boundaries of the
sample where orbital magnetoresistance is absent. This phe-
nomenon was dubbed 'static skin effect' in analogy with the skin
effect in metals. The latter refers to the fact that the density of an

alternating current (AC) is largest near the surface of the con-
ductor and decreases exponentially with increasing depth. Note,
however, that the conductivity profile in the 'static' version of the
skin effect has a completely different origin.

The large magnetoresistance of bismuth can be understood in
the semiclassical picture of cyclotron orbits shrinking with
increasing magnetic field (Fig. 6a). When the magnetic field is
parallel to a surface, within a thickness of the order of cyclotron
radius, electrons can conduct much better than in the bulk and
generate a sizeable contribution to the total conductivity (Fig. 6b).

However, this semiclassical scenario fails to explain two key
observations. The first is the fact that, as we saw above, the
amplitude of the effect is unchanged when the magnetic field is
changed by a factor of 50 (see Fig. 5). This is puzzling in the
'static skin effect' scenario where the distribution of current
depends on the ratio of the cyclotron radius and the effective
sample thickness[30]. As illustrated in Fig. 6c, increasing the
magnetic field will reduce the width of the cyclotron edge and will

enhance the difference in the conductivity of the bulk *vs.* edge. One may argue that these two tendencies may approximately cancel out generating an additional conductivity which does not vary much with the amplitude of the field. However, a perfect cancellation would be mysterious given the difference in the evolution of the magnetoresistance and the cyclotron radius. In addition, how to explain the indifference of the angular width of the peak to the magnitude of the magnetic field? The observation implies that the current profile has remained the same despite a 50-fold shrink in the size of the cyclotron radius.

The second observation is the absence of this phenomenon in Sb. One may be tempted to invoke a possible difference in surface rugosity. However, there is no evidence for such a difference. In addition, it is unlikely for a quantitative difference in surface quality to totally erase the effect and make the outcome qualitatively different.

The contrast between Bi and Sb can be traced to a fundamental feature of their electronic properties involving the symmetry of their band structure.

**Band inversion and topological invariants: Bi *vs.* Sb**. The third-neighbor tight-binding model conceived by Liu and Allen[6] gives a successful account of the electronic band structure of bulk bismuth and antimony, as documented by numerous experiments. The model quantifies hopping energies between first, second, and third neighbors with unprimed ($V$), primed ($V'$) and double-primed ($V''$) parameters, respectively. The crystal lattice has two sub-lattices, i.e., the unit cell includes two atoms. The third neighbor of the original atom is the closest neighbor on the same sub-lattice and both atoms reside in the same monolayer. The three first and the second neighbor atoms belong to a different sub-lattice and lie in other monolayers above and below the original atom (see Fig. 2a). The 14 adjustable parameters of the model were chosen to give the best agreement with experiment. An additional parameter was spin–orbit coupling (SOC), $\lambda$, which was taken to be 0.6 eV for Sb and 1.5 eV for Bi. This model gives a

reasonable account of the Fermi surface pockets of electrons and holes and the direct and the indirect gaps of Bi and Sb[6].

In 2007, $Bi_{1-x}Sb_x$ alloys were identified as the first bulk topological insulators[17,18,46], based on an important difference between Sb and Bi band symmetries. The starting point of this identification was the band inversion at the high-symmetry $L$-point in the bulk Brillouin zone, found in this tight-binding model, as well as in previous works[47,48]. The symmetry of the wave function at the $L$-point can be classified into symmetric ($L_s$) and antisymmetric ($L_a$) with respect to space inversion, where the eigenvalues of parity operator are $+1$ for $L_s$ and $-1$ for $L_a$. As one can see in panels a and b of Fig. 7, in bismuth the conduction band at the $L$-point is symmetric and the valence band is antisymmetric, while the inverse is true in the case of antimony.

There are 8 high-symmetry (one $\Gamma$, one $T$, three $X$, and three $L$) points in the Brillouin zone (Fig. 7c). They remain invariant under inversion and time-reversal operators. At $\Gamma$-, $T$-, and $X$-points, there is no difference in parity invariants between Sb and Bi. On the other hand, there is one for the $L$-points. Kane and collaborators argued that the difference in parity invariants at the $L$-points leads to a $Z_2$ topological invariant dichotomy between the two systems. As a result, at zero magnetic field, topology of the system is trivial in Bi and non-trivial in Sb[18].

Let us briefly discuss what drives this band inversion. Both Bi and Sb crystallize in the A7 rhombohedral crystal structure, which can be assimilated to an assembly of two distorted FCC sub-lattices. As seen in Table 2, there is a significant difference between tight-binding parameters of Bi and Sb. $V_{pp\sigma}$ is the hopping energy of sigma bonding of $p$-orbitals of the first neighbors and $V'_{pp\sigma}$ is the same quantity for second neighbors. One can see that their relative difference is much larger in Sb than in Bi[6]. As a consequence, the Peierls gap is larger in Sb than in Bi. The larger gap hinders the band crossing and the reversal of $L_s/L_a$ hierarchy.

In a conventional Peierls transition, the energy of the symmetric band is lower than that of the antisymmetric band[49,50]. (An intuitive picture of this hierarchy is sketched in

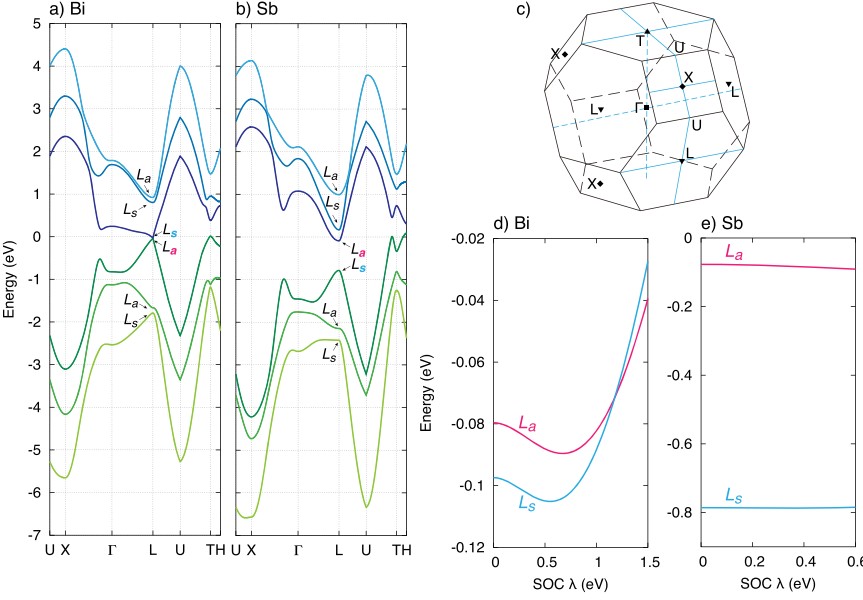

**Fig. 7 Band inversion due to spin–orbit coupling.** Band structure of **a** Bi and **b** Sb. In Bi, the gap between conduction and valence bands at the $L$-point is small and the hierarchy between them is inverted from the ordinary hierarchy under the Peierls transition. The upper band is the symmetric $L_s$ and the lower is the antisymmetric $L_a$. **c** The Brillouin zone and its high-symmetry points. **d**, **e** Energies of $L_s$ and $L_a$ as a function of the magnitude of SOC for Bi and Sb. The effective $\lambda$ for Bi and Sb yielding the best fit to experiment are mentioned in Table 2. The energy hierarchy $L_s/L_a$ is altered by SOC in Bi, whereas it is not in Sb. All these calculations were carried out by using the Liu-Allen's tight-binding model[6].

**Table 2 A comparison of bismuth and antimony.**

| Parameter | Bi | Sb |
|---|---|---|
| $d_1$ (Å) | 3.5120 | 3.3427 |
| $d_2$ (Å) | 3.0624 | 2.9024 |
| $\mu$ | 0.2341 | 0.2336 |
| $\alpha$ | 57° 19' | 57° 14' |
| $V_{pp\sigma}$ (eV) | 1.854 | 2.342 |
| $V'_{pp\sigma}$ (eV) | 1.396 | 1.418 |
| $V_{pp\pi}$ (eV) | −0.600 | −0.582 |
| $V'_{pp\pi}$ (eV) | −0.344 | −0.393 |
| $V/V'(\sigma)$ | 1.33 | 1.65 |
| $V/V'(\pi)$ | 1.74 | 1.48 |
| $\lambda$ (eV) | 1.5 | 0.6 |

The nearest-neighbor distance, $d_1$, and the second nearest-neighbor distance, $d_2$, in Bi and Sb. They are longer in Bi where atoms are larger. But, the relative distance between the two sub-lattices, $\mu$ and the rhombohedral angle, $\alpha$ are almost the same. On the other hand, the tight-binding parameters in Sb and in Bi are different[6].

the next subsection.) This is exactly what happens in Sb. However, the SOC can alter this energy hierarchy. In Fig. 7d and e, we plot the energies of the conduction and valence bands at the $L$-point for Bi and Sb as a function of the magnitude of SOC using the Liu-Allen model. The conduction and valence bands of Sb are hardly affected and the energy hierarchy is unchanged by the SOC. On the other hand, the hierarchy is inverted by the SOC for Bi. This hierarchy alternation happens only in Bi, because the band gap (i.e., the lattice distortion) is much smaller and the SOC is larger than in Sb. Actually, the bands of Sb would be inverted if the SOC were unrealistically large (~20 eV).

Interestingly, the $L_s$ and $L_a$ bands are distinguished by the parity inversion[18,51].

**Parity and symmetry of the wave functions**. The gap opening at the $L$-point originates from the Peierls distortion[3,52]. The real-space image of the wave functions of conduction and valence bands are depicted in Fig. 8 [49,50]. Peierls distortion can be understood as a dimerization, where the lattice is distorted to generate pairs as seen in Fig. 8a. (Each dimer corresponds to a unit cell of Bi or Sb. Two atoms in the dimers are the first nearest neighbor with each other (Fig. 2a). There are three Peierls chains crossing at each atom in the rhombohedral structure.) The wave functions of a single dimer are given in terms of the bonding and anti-bonding orbitals. The wave function of the symmetric band ($\psi_s$) is given by the periodic array of bonding orbitals, whereas the wave function of the antisymmetric band ($\psi_a$) is given by that of anti-bonding orbitals (Fig. 8b). It is clear from Fig. 8b that, in absence of dimerization, the energy of $\psi_s$ is degenerate with that of $\psi_a$. (If one removes the dashed boxes from Fig. 8b, one finds that the two wave functions are equivalent in an infinite system.) Dimerization lowers the energy of $\psi_s$ compared to $\psi_a$, because the energy of bonding orbitals should be lower than that of anti-bonding orbitals. The magnitude of the energy gap between $\psi_s$ and $\psi_a$ is determined by the degree of dimerization, which is roughly given by the difference between the intra- and inter-dimer hopping, i.e., the difference between $V_{pp\sigma}$ and $V'_{pp\sigma}$. It is evident from Fig. 8b that $\psi_s$ is symmetric and $\psi_a$ is antisymmetric for the space inversion, where the inversion center of the crystal locates at the bond center in the dimer.

Now, let us consider the reflection of cyclotron orbits at the boundary with these wave functions. We only consider specular reflection normal to the surface for the sake of simplicity. (Although the incidence angle is not restricted to be normal in general, normal reflection is expected to play a major role.) By the normal reflection, the wave vector of electrons changes as $\boldsymbol{k}$ to $-\boldsymbol{k}$,

which corresponds to the parity operation $P$. The parity operation results in $P\psi_s = +\psi_s$ and $P\psi_a = -\psi_a$[18,47,48]. The sign of the antisymmetric wave function is changed by the reflection at the boundary (Fig. 8) only for $\psi_a$. Therefore, naively, one expects a qualitative difference in boundary reflection between $\psi_s$ and $\psi_a$. Further theoretical investigations are required to shed light on this subject.

**Electron topology at the cyclotron edge**. The static skin effect picture[29,30] is a semiclassical approach which does not take into account the phase of the electrons' wave function. The interruption of cyclotron orbits was framed in a specular-diffusive dichotomy. If the reflection is perfectly specular, momentum is conserved and there is an additional contribution to conductivity. The effect will weaken if the reflection becomes partially diffusive. However, reflected electron waves can interfere with incoming waves. The electronic Fabry interferometers employed in two-dimensional electron gases[53] are an eloquent demonstration of this fact.

In a quantum treatment of the interruption of the cyclotron orbits by sample boundaries, it is crucial to consider the fate of the electron wave function and its phase following a mirror transformation. As we saw above, quasi-particles residing in the electron pockets have opposite symmetries in Bi and in Sb. The symmetric conduction band of bismuth, in contrast to the antisymmetric band of Sb, allows a constructive interference upon a mirror reflection at the boundary. However, it remains to be seen how this difference survives in presence of quantizing magnetic field.

Even in the simple isotropic case, large-index Landau wave functions have a non-trivial angular distribution in real space[54], which is to be affected by the zero-field anisotropy of the Fermi momentum. The anisotropic cyclotron orbits of Bi surface states have recently become accessible to experiment, thanks to scanning tunneling microscopy studies[12].

It is tempting to draw an analogy between the present context and Floquet systems in which topological protection is provided by a periodic perturbation[31–33]. At zero magnetic field, the electronic surface states of a solid are distinct from bulk states by the abrupt interruption of the lattice. There is another distinction, which emerges at high magnetic field. The surface states are periodically disturbed by cyclotron orbits of the bulk. This leads to a spatio-temporal discrete translation symmetry[55,56]. An edge atom at a given position and time is not instantaneously equivalent to its neighbor. One of the two may be perturbed by an electron from the bulk in its cyclotron orbit, in contrast to the other. On the other hand, the two atoms remain equivalent if the temporal periodicity is taken into account. In other words, the discrete symmetries of space and time become intertwined: $(\overrightarrow{r}, t) \rightarrow (\overrightarrow{r} + \overrightarrow{a}, t + \frac{2\pi}{\omega_c})$. Future studies will tell if this analogy plays any role in explaining our observation.

One of the motivations of the present study was the theoretical proposal that bismuth is a higher-order non-trivial topological system[23]. This was put forward to explain the origin of ballistic transport in Bi nanowires detected by superconducting proximity studies[28]. Note that the boundary ballistic transport detected in our experiment is two-dimensional and does not appear to arise from one-dimensional channels expected in the case of topologically protected hinge states[23].

In conclusion, we found that in bulk crystals of bismuth, there is a robust contribution to conductance when the magnetic field is aligned parallel to a two-dimensional boundary of the sample. The absence of this effect in antimony implies that the difference in symmetry of the conduction band has a significant outcome.

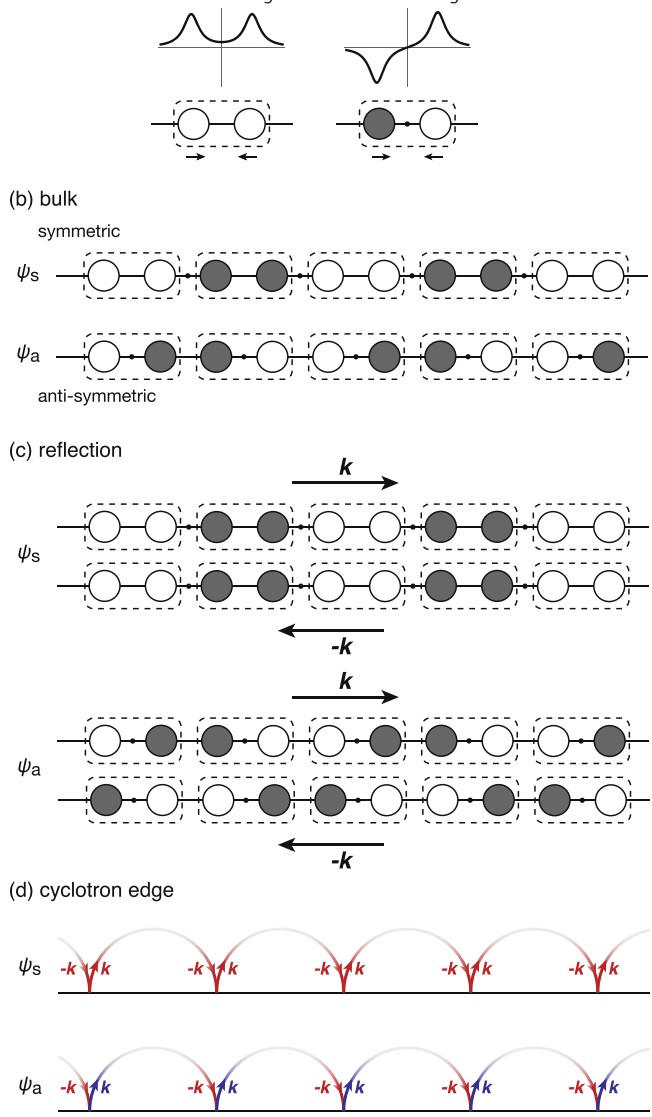

**Fig. 8 Wave functions of electrons with the Peierls distortion. a** The wave function in the unit cell (dimer) can be expressed in terms of bonding and anti-bonding molecular orbits. The white and black circles express the sign of the wave functions. The dots express the positions of nodes in the wave function. **b** The symmetric ($\psi_s$) and antisymmetric ($\psi_a$) wave functions can be expressed in terms of bonding and anti-bonding as well. **c** The sign of $\psi_a$ is inverted by the parity operation ($k \rightarrow -k$), while that of $\psi_s$ is unchanged. It is thus naively expected that $\psi_a$ is strongly disturbed around the boundary by the normal reflection. **d** Illustration of the parity operation in the cyclotron edge. In the antisymmetric case, there is a $\pi$ phase shift between incoming and reflected cyclotron orbits interrupted by the boundary.

A satisfactory explanation of our results is missing and remains a challenge for theory. While the 'static skin effect' explains the existence of boundary conductance in a macroscopic crystal, it fails to explain its absence in antimony as well as the robust behavior of the conducting channel in presence of strong magnetic fields. We note that the semiclassical 'static skin effect' has not yet been formulated in a quantum-mechanical frame incorporating the known contrast between the parity of the Bloch waves in Bi and in Sb at the L-point. We argued that the latter may affect reflection at the crystal boundaries. To the best of our knowledge, this has not yet been addressed by theory.

Our result has implications for several puzzling observations previously reported in bismuth. The loss of threefold symmetry in transport measurements[9,10] finds a natural explanation. It may also be invoked to explain the loss of symmetry seen by thermodynamic probes[11]. The anomaly caused by the evacuation of a Landau level is a van-Hove singularity with a cut-off due to disorder and finite size. The latter correlates with the shape of the sample. This would imply that the finite size cut-off of the van-Hove singularity may be different for different field orientations. Thermodynamic measurements on samples with different shapes and different sizes will be instructive to check this. Finally, our observation may indicate that boundaries of a bismuth crystal in presence of magnetic field provide a topological barrier. This would provide a possible solution to the puzzle of distinct chemical potentials between twinned crystals of bismuth[7].

## Methods
Bi and Sb crystals were commercially obtained through MaTecK GmbH, which oriented and cut them to the desired shape and dimensions. The sample surface was not polished and did not go through any other specific treatment. The experiments were performed in two different locations (Ewha University and ESPCI) and with two different set-ups. A home-made set-up was used in Ewha University and a Quantum Design PPMS was used in ESPCI Paris. In both cases, resistivity was measured with a standard 4-wire configuration and electrical contacts were made with silver paste. In order to ensure the homogeneity of charge flow, current electrodes were applied to a thick layer of silver paste (Dupont 4929N) covering most of the end surfaces of the prism except for a small circular area in the center. Then a small island of silver paste was created in the center and voltage electrodes were placed on it. At Ewha University, magnetoresistance was measured with an AC method with a typical current of 100 μA at frequencies between 13 and 17 Hz. For resistivity measurements as a function of temperature, a DC method was used with a typical current of 1 mA. In Paris, measurements were performed using currents between 1 mA and 5 mA and the resistivity option of the PPMS in AC drive mode (50 Hz square wave excitation).

## Data availability
All data generating the plots presented in this study can be obtained from the corresponding author upon request.

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

## Acknowledgements

This work was supported by the Agence Nationale de la Recherche (ANR-18-CE92-0020-01 and ANR-19-CE30-0014-04), by Jeunes Equipes de l'Institut de Physique du Collège de France and by a grant attributed by the Ile de France regional council. W.K. was supported by the National Research Foundation of Korea (NRF) (No. 2018R1D1A1B07050087 and No. 2018R1A6A1A03025340).

## Author contributions

W.K. and K.B. designed the experiments. W.K and F.S. carried out the experiments. Y.F. performed theoretical calculations. F.S., B.F., W.K. and K.B. analyzed the data. Y.F. and K.B. wrote the manuscript with input from all authors.

## Competing interests

The authors declare no competing interests.
