## [Peer Review File · Nature Communications]

REVIEWER COMMENTS

Reviewer #1 (Remarks to the Author):

The work presented in this manuscript highlights the effect of bismuth's band structure topology on its edge-bulk conductivity in the high magnetic field limit, amongst others providing an explanation for the very large magnetoresistance in samples of very high crystallographic quality.

The overall quality of the experimental work and (sometimes 'handwaving') interpretation is good but currently seems to lack a 'killer selling point' that is of great interest to the wide audience of Nature Communications readers. This is based on two observations, first, the authors claim that the present manuscript solves various existing open questions (e.g. loss of rotational threefold symmetry, source of topological barrier), but the majority of those questions come from their own previous work.

Whereas that might merely display the community's gradual progress in fundamental understanding, it somehow also shows that few other research groups seem to be currently working on this topic.

Many of this work's references are either classic and related to fundamental bismuth-research from the last century or are co-authored by one or more of this manuscript's co-authors (at least 11 references).

Can the authors make clearer how this work is of interest to the broader research community, e.g. what are the (evidenced) implications for related research topics and/or materials? Or put bluntly, who will cite this work and for what reasons? Potential literature sources could come from the statement in Section III.B "the electronic band structure of bulk bismuth and antimony, as documented by numerous experiments", which is currently unreferenced.

Secondly, and related to this, various sections of the theoretical background/interpretation of the experimental results are 'handwaving' (e.g. regarding the "static skin effect" and "Peierls distortion") and lack a certain depth to be able to be of direct use to others. Whereas the overall story is consistent, and the apparent simplicity of some of the argumentation and analogies will appeal to the non-expert reader, it would be of great benefit to the expert reader if these sections could be expanded with details that are now left as a "challenge for future theoretical studies/investigations". Most detail is currently provided on the band structure calculations of Bi and Sb and the associated parameters, but this is fully reproduced (as stated and referenced) from previous work of Liu and Allen. The manuscript would be greatly enhanced if the authors contributed to additional quantitative analysis of the presented experimental results.

To assist potential academic reproduction of the work, the expert reader would also benefit from a greater level of detail in the experimental description as well as the analysis and theoretical interpretation, which could either be added to the main text or to the supplemental material. How were the resistance measurements carried out? AC/DC, excitation current strengths, contact configuration, and how were the sample contacts established? Why should the extrema of the excess conductivity r follow a cosine-behaviour? What is the physical meaning of the free-fitting parameter q ? A final criticism is the lack of detail on the sample dependence. As highlighted in Table 1, RRR and ρ_0 vary substantially between the presented samples, roughly with a factor of two. Most of the results, however, have been obtained using the 'relatively lowest crystallographic quality' sample 1a. For measurements at lowest temperatures, one might expect a richer/more pronounced angle-dependent magnetoresistance profile for a 'higher quality' sample, which would also provide a fairer comparison to an 'equal quality' sample with a different face orientation. In turn, this might have a direct impact on any features within Figure 5a and b.

A few minor comments:

- In Figure 1, the caption describes a decreasing conductivity, but increasing resistivity with magnetic field is plotted. Although this might be equivalent, it does confuse. Furthermore, for Bi the deviation from B^2 behaviour appears to be at high magnetic field, not at the stated low end.
- Can the authors elaborate on the statement in Section II.A "The latter was dubbed 'extremely large magnetoresistance' by many authors."?
- What is the purpose of the paragraph in Section II.C "One question to address is how large the magnetic field requires to be. Does this phenomenon emerge by approaching the quantum limit (which corresponds to confining electrons to their lowest Landau levels)? In this case, the criterion

would be a magnetic field at which the cyclotron energy becomes comparable to the Fermi energy. Or does it emerge as soon as the high-field limit is attained? The latter corresponds to a cyclotron radius short enough to be completed without scattering. These two field scales differ significantly. To confine all electrons of one pocket to their lowest Landau level, a magnetic field of 1.3 T to 1.6 T is needed in the binary-bisectrix plane. On the other hand, quantum oscillations are observable at fields as small as 0.1 T." if most of this is then repeated into the subsequent paragraphs?

- Can the authors support the statement "One can also see that when distinct, the two type of peaks have different angular width and slightly different amplitudes" with 'typical' values?
- Can the authors describe how the colour plot of Figure 5a was made? At which static fields were the angle-dependent magnetoresistance measurements carried out, and what type of interpolation method was used to generate the areas in-between them?
- Where is Section III of the Supplemental Material referred to in the main text?
- Various spelling corrections can be made while the authors adapt and reread their manuscript, e.g. "mobilityess", "magneto-conductivity", "magnetoconductivity", "Fig. ???" (also in Supplemental Material), "two type of peaks", etc.

Reviewer #2 (Remarks to the Author):

In the manuscript "Boundary conductance protected by topology in macroscopic bismuth crystals," W. Kang et al. reported the bulk angular-dependence of magnetoresistance that depends on the surface orientation, and also found that such an effect is absent in antimony, which has the same crystal structure and the similar electronic structure as Bi. The authors attributed this effect to the parity difference of the Bloch wave in Bi and Sb.

In my opinion, the work contains solid, intriguing and important results. It is generally believed that for bulk samples, the angle-dependent magnetoresistance would be independent of the surface conditions and would depend only on the symmetry of the crystal (as shown in the authors' previous work on Bi in ref 10, 12). In this study, the authors have shown that this assumption is incorrect and that the macroscopic shape of the sample surface has a significant effect on the angle-dependent magnetoresistance. Similar attempts to discuss the electronic nematicity from the in-plane angular dependence of magnetoresistance have been made in various systems including doped Bi₂Se₃ and CsV₃Sb₅. The present results are significant because they urge a reconsideration of the interpretation of previous experiments. Therefore, I think that the manuscript is suitable for serious consideration of publication in Nature Communications, provided that the authors address the following issues.

- (1) One of the main points of this paper is the surface dependence of the conductivity of bismuth. Therefore, it is important to describe the surface treatment method of bismuth. Each sample is supposed to have polished surfaces, but it is desired to describe whether the surface is mirror-like or rough, or whether the flatness of the surface affects the surface conductivity. It would also be beneficial to add the method section, in which other experimental configurations are described as well.
- (2) The authors argue that "the finite size cut-off of the van-Hove singularity may be different for different field orientations" and the surface-dependent magnetotransport may "explain the loss of symmetry seen by thermodynamic probes." In this case, what is important is the ratio of surface conductivity to bulk conductivity. If surface conductivity exists, it is naively assumed that the larger the bulk sample size, the smaller the contribution of surface conductivity. In particular, for thermodynamic measurements, the effect of surface conditions is generally considered to be small. Can the authors answer the question about the effect of sample size on the angular dependence of magnetoresistance? Also, it would be meaningful to describe the size of the voltage contacts in Fig. 2.
- (3) Although the title of the paper includes the words "protected by topology," my understanding is that the experimental results do not directly imply the topological nature of the material. Theoretically, the authors suggest that the difference in parity of the wave function may contribute to the difference between bismuth and antimony. However, there is also no direct experimental evidence to support this, and since the authors mention that "at zero magnetic field, topology of the system is trivial in Bi

and non-trivial in S_b ," it does not seem appropriate to use the term topological. The authors should therefore explain the meaning of the word topological in the title of the paper.

Here are minor corrections.

- (1) "mobilityess" in section IIA.
- (2) Fig. ?? (Fig number not indicated) in section IIC.
- (3) The paragraph in the middle of the right column on page 5 ends with a comma.
- (4) "... the cyclotron edge does NOR evolve..." in the caption of Fig. 6.
- (5) No closing parenthesis after mentioning Fig. 8 in the right bottom corner of page 10.

Reviewer #1 (Remarks to the Author):

The work presented in this manuscript highlights the effect of bismuth's band structure topology on its edge-bulk conductivity in the high magnetic field limit, amongst others providing an explanation for the very large magnetoresistance in samples of very high crystallographic quality.

The overall quality of the experimental work and (sometimes 'handwaving') interpretation is good,

We thank the reviewer for this generous comment. We need to specify that while our experiment detects edge conductivity in quantizing magnetic fields, it does NOT "provide an explanation for the very large magnetoresistance in samples of very high crystallographic quality." The large magnetoresistance of bismuth crystals is a bulk property due to the ultra-high mobility of the charge carriers (as high as $10^8 \text{ cm}^2\text{s}^{-1}\text{V}^{-1}$).

but currently seems to lack a 'killer selling point' that is of great interest to the wide audience of Nature Communications readers.

This is based on two observations, first, the authors claim that the present manuscript solves various existing open questions (e.g. loss of rotational threefold symmetry, source of topological barrier), but the majority of those questions come from their own previous work. Whereas that might merely display the community's gradual progress in fundamental understanding, it somehow also shows that few other research groups seem to be currently working on this topic. Many of this work's references are either classic and related to fundamental bismuth-research from the last century or are co-authored by one or more of this manuscript's co-authors (at least 11 references). Can the authors make clearer how this work is of interest to the broader research community, e.g. what are the (evidenced) implications for related research topics and/or materials? Or put bluntly, who will cite this work and for what reasons? Potential literature sources could come from the statement in Section III.B "the electronic band structure of bulk bismuth and antimony, as documented by numerous experiments", which is currently unreferenced.

As the reviewer correctly points out, two open questions are addressed by our new findings: i) The origin of the loss of rotational symmetry ('nematicity'), and ii) the electronic topology. Regarding the first question, it is true that we have devoted several papers to this issue. However, we are not alone. As an example of other studies of nematicity in bismuth, see Science 354, 316 (2016). As for the second question, it is investigated by many groups. The paper entitled "Higher-order topology in bismuth", Nature physics 14, 918 (2018) has been cited 407 times (source: Google Scholar, 14/09/2021). These facts may not provide a satisfactory answer to a question like "who will cite this work?". However, they confirm that investigating bismuth in 2021 is not like speaking in an empty room or pushing open doors. We are addressing genuine puzzles of considerable interest to members of our community. In this context, let us note that reviewer 2 states that "The present results are significant because they urge a reconsideration of the interpretation of previous experiments" and pointed out the possible relevance of the results to doped Bi_2Se_3 and CsV_3Sb_5 .

The immediate implications of our work for these two issues are the following: i) The mysterious loss of rotational symmetry (observed in numerous bismuth crystals of different shapes) is not a consequence of 'valley nematicity' as speculated before but is due to the presence of conducting edge modes; ii) The band symmetry at L-point, of opposite signs in bismuth and antimony, leads to a qualitatively different behavior at high magnetic fields. This means that the trivial/non-trivial

dichotomy, which has attracted tremendous attention in the last decade is not necessarily the same at zero field and in presence of quantizing magnetic fields.

It is worth to mention that the debate on the topology of the surface states in bismuth has generated a voluminous literature in recent years and the trivial or non-trivial debate is still going on (see arXiv:2108.12674 for a recent contribution to this debate). Following the reviewer's recommendation, we have added a paragraph and several new references in the introduction informing the reader about this.

Secondly, and related to this, various sections of the theoretical background/interpretation of the experimental results are 'handwaving' (e.g. regarding the "static skin effect" and "Peierls distortion") and lack a certain depth to be able to be of direct use to others. Whereas the overall story is consistent, and the apparent simplicity of some of the argumentation and analogies will appeal to the non-expert reader, it would be of great benefit to the expert reader if these sections could be expanded with details that are now left as a "challenge for future theoretical studies/investigations". Most detail is currently provided on the band structure calculations of Bi and Sb and the associated parameters, but this is fully reproduced (as stated and referenced) from previous work of Liu and Allen. The manuscript would be greatly enhanced if the authors contributed to additional quantitative analysis of the presented experimental results.

The two concepts mentioned by the reviewer, the "static skin effect" and "Peierls transition" are old ideas proposed decades ago. The first is Azbel's idea that cyclotron motion of electrons interrupted at the boundaries would not generate magnetoresistance. The second refers to the notion that the rhombohedral structure in bismuth and antimony is a case of Peierls transition. It is true that they are "forgotten" by many contemporary researchers, but why should they be considered as "handwaving"? Azbel's idea led to the discovery of cyclotron resonance in metals (Journal of Physics and Chemistry of Solids 6, 113 (1958)). Analyzing the crystal structure of bismuth as a case of "Peierls transition" was an idea proposed in 1930s (R. Peierls, "More surprises in theoretical Physics", page 24). Our source for its application to the present problem comes from a Chemistry textbook (R. Hoffmann, "Solids and Surfaces: A Chemist's View of Bonding in Extended Structures"). The fact that these concepts are unfamiliar to many practicing condensed-matter physicists does not diminish their depth.

This said, there may be a valid point in the reviewer's remark. These ideas are not sufficient to explain our experimental observation and merely constitute starting points. In the present version, we highlight that our observation remains unexplained and stress the fact that this is because of the absence of a rigorous theory of the consequences of band symmetry for cyclotron motion of electrons near the boundary. This will hopefully motivate future theoretical efforts.

To assist potential academic reproduction of the work, the expert reader would also benefit from a greater level of detail in the experimental description as well as the analysis and theoretical interpretation, which could either be added to the main text or to the supplemental material. How were the resistance measurements carried out? AC/DC, excitation current strengths, contact configuration, and how were the sample contacts established? Why should the extrema of the excess conductivity r follow a cosine-behaviour? What is the physical meaning of the free-fitting parameter q ?

Our experiments are extremely simple. What can be more standard than measuring the resistivity of large commercial single crystals with a PPMS? This is the reason behind the absence of detailed experimental specification. We are confident about reproducibility since the experiments were carried out in two different locations and with two different set-ups. In the new version, we insert a section entitled "Mmethods". We have also added a couple of sentences in section III.A specifying the

significance of the parameters used to fit the excess of conductivity and answering the reviewers' questions.

A final criticism is the lack of detail on the sample dependence. As highlighted in Table 1, RRR and ρ_0 vary substantially between the presented samples, roughly with a factor of two. Most of the results, however, have been obtained using the 'relatively lowest crystallographic quality' sample 1a. For measurements at lowest temperatures, one might expect a richer/more pronounced angle-dependent magnetoresistance profile for a 'higher quality' sample, which would also provide a fairer comparison to an 'equal quality' sample with a different face orientation. In turn, this might have a direct impact on any features within Figure 5a and b.

The short answer to this question is that the main claim of our paper, which is shown in Fig. 4 does not display any sample dependence. All four crystals show the same behavior: namely when the magnetic field is parallel to a surface there is a peak in conductivity (blue arrows). These peaks sometimes add up to the peaks in conductivity due to the bulk magnetoconductivity (red arrows). Besides this difference (caused by sample geometry), there is no detectable difference. The same scale has been chosen for the four panels and the lower background in the case of panel d can be traced to its higher mobility (and therefore higher magnetoresistance).

Note also that the analysis presented in Fig.5 can only be done with two triangular prisms whose edges are parallel to C1 and C2. It happens that the mobility in the four triangular samples was lower than in the two cubic samples, but there is no reason to believe that a twofold difference in mobility has a consequence for this physics. During the past decade, we have studied numerous bismuth crystals of different qualities and geometries and in all of them the rotational symmetry was lost at low temperature and moderate magnetic field.

A few minor comments:

- In Figure 1, the caption describes a decreasing conductivity, but increasing resistivity with magnetic field is plotted. Although this might be equivalent, it does confuse. Furthermore, for Bi the deviation from B^2 behaviour appears to be at high magnetic field, not at the stated low end.

Thank you! Corrected.

- Can the authors elaborate on the statement in Section II.A "The latter was dubbed 'extremely large magnetoresistance' by many authors."?

A large non-saturating magnetoresistance was discovered in bismuth by Kapitza in 1928. However, in the first decades of the 21st century, the fact this large orbital magnetoresistance is a generic feature of low-density semi-metal has been forgotten. This orbital magnetoresistance easily overwhelms what was called 'giant' or 'colossal' magnetoresistance, which involve the spin degree of freedom and was discovered in the late 20th century with technological implications. We have added a sentence and a couple of references to put the buzzword in its true context.

- What is the purpose of the paragraph in Section II.C "One question to address is how large the magnetic field requires to be. Does this phenomenon emerge by approaching the quantum limit (which corresponds to confining electrons to their lowest Landau levels)? In this case, the criterion would be a magnetic field at which the cyclotron energy becomes comparable to the Fermi energy. Or does it emerge as soon as the high-field limit is attained? The latter corresponds to a cyclotron radius short enough to be completed without scattering. These two field scales differ significantly. To confine all electrons of one pocket to their lowest Landau level, a magnetic field of 1.3 T to 1.6 T is needed in the

binary-bisectrix plane. On the other hand, quantum oscillations are observable at fields as small as 0.1 T." if most of this is then repeated into the subsequent paragraphs?

Thank you for noticing this. We have rewritten this section and suppressed the repetition.

- Can the authors support the statement "One can also see that when distinct, the two type of peaks have different angular width and slightly different amplitudes" with 'typical' values?

Yes! We have added 'typical values'

- Can the authors describe how the colour plot of Figure 5a was made? At which static fields were the angle-dependent magnetoresistance measurements carried out, and what type of interpolation method was used to generate the areas in-between them?

Added.

- Where is Section III of the Supplemental Material referred to in the main text?

Corrected.

- Various spelling corrections can be made while the authors adapt and reread their manuscript, e.g. "mobility", "magneto-conductivity", "magnetoconductivity", "Fig. ??", (also in Supplemental Material), "two type of peaks", etc.

Thanks! Corrected!

Reviewer #2 (Remarks to the Author):

In the manuscript "Boundary conductance protected by topology in macroscopic bismuth crystals," W. Kang et al. reported the bulk angular-dependence of magnetoresistance that depends on the surface orientation, and also found that such an effect is absent in antimony, which has the same crystal structure and the similar electronic structure as Bi. The authors attributed this effect to the parity difference of the Bloch wave in Bi and Sb.

In my opinion, the work contains solid, intriguing and important results. It is generally believed that for bulk samples, the angle-dependent magnetoresistance would be independent of the surface conditions and would depend only on the symmetry of the crystal (as shown in the authors' previous work on Bi in ref 10, 12). In this study, the authors have shown that this assumption is incorrect and that the macroscopic shape of the sample surface has a significant effect on the angle-dependent magnetoresistance. Similar attempts to discuss the electronic nematicity from the in-plane angular dependence of magnetoresistance have been made in various systems including doped Bi₂Se₃ and CsV₃Sb₅. The present results are significant because they urge a reconsideration of the interpretation of previous experiments. Therefore, I think that the manuscript is suitable for serious consideration of publication in Nature Communications, provided that the authors address the following issues.

We thank the reviewer for this assessment.

(1) One of the main points of this paper is the surface dependence of the conductivity of bismuth. Therefore, it is important to describe the surface treatment method of bismuth. Each sample is supposed to have polished surfaces, but it is desired to describe whether the surface is mirror-like or rough, or whether the flatness of the surface affects the surface conductivity. It would also be beneficial to add the method section, in which other experimental configurations are described as well.

The surfaces of our samples were not polished and did not go through any other additional treatment. The samples were measured as cut. Photographs of a pair of samples can be seen in Fig. 1c. We have followed the reviewer's recommendation and have added a method section specifying these details.

(2) The authors argue that "the finite size cut-off of the van-Hove singularity may be different for different field orientations" and the surface-dependent magnetotransport may "explain the loss of symmetry seen by thermodynamic probes." In this case, what is important is the ratio of surface conductivity to bulk conductivity. If surface conductivity exists, it is naively assumed that the larger the bulk sample size, the smaller the contribution of surface conductivity. In particular, for thermodynamic measurements, the effect of surface conditions is generally considered to be small. Can the authors answer the question about the effect of sample size on the angular dependence of magnetoresistance? Also, it would be meaningful to describe the size of the voltage contacts in Fig. 2.

We concede that the proposed link between our transport data and the interpretation of the thermodynamic (magnetostriction) data is speculative and this is the reason for the presence of the word "may" in the above-cited sentence. Presently we have no data on size dependence, and we cannot therefore address the reviewer's question. The study of the size dependence is an interesting idea for future investigation. In the new version, we provide specifications on the voltage and current contacts in the caption of Fig.2 as well as in the new section on methods.

(3) Although the title of the paper includes the words "protected by topology," my understanding is that the experimental results do not directly imply the topological nature of the material. Theoretically, the authors suggest that the difference in parity of the wave function may contribute to the difference between bismuth and antimony. However, there is also no direct experimental evidence to support this, and since the authors mention that "at zero magnetic field, topology of the system is trivial in Bi and non-trivial in Sb," it does not seem appropriate to use the term topological. The authors should therefore explain the meaning of the word topological in the title of the paper.

The reviewer is entirely right. Following this criticism and in order to be as accurate as possible, we have modified the title of the paper, which now reads "Boundary conductance protected by band symmetry in macroscopic bismuth crystals". In contemporary scientific literature, the word topology has a very specific meaning. What we observe is an effect present in a symmetric conduction band and absent in an antisymmetric conduction band. We believe that this new title summarizes the main finding without misleading the reader about the context.

Here are minor corrections.

- (1) "mobilities" in section IIA.
- (2) Fig. ?? (Fig number not indicated) in section IIC.
- (3) The paragraph in the middle of the right column on page 5 ends with a comma.
- (4) "... the cyclotron edge does NOR evolve..." in the caption of Fig. 6.
- (5) No closing parenthesis after mentioning Fig. 8 in the right bottom corner of page 10.

Thanks! They are all corrected!

REVIEWER COMMENTS

Reviewer #1 (Remarks to the Author):

As expressed in the previous reviewing round, I think that the authors' scientific work is worth publishing and reading, and has several implications for existing research questions and a high likelihood to lead to follow-up research. Nevertheless, I still struggle with both the current presentation and interpretation of the work, to which the authors have only made very few changes with respect to their original manuscript.

Most statements made throughout the manuscript are scientifically supported by the authors' previous publications, not by those of others. As I highlighted previously, and now rephrased in a very polarised fashion for emphasis purposes: either the presented work is relevant to a large scientific audience and the fundamental academic work that it builds upon is cited at crucial stages throughout the manuscript, or this work merely builds on the authors' previous findings, which, although well-read, are only of interest to a small sub-set of this journal's readers. I am fully aware that eventually my previous statements only lead to a discussion based on subjective, research-ethical arguments, that might go beyond the scope of a manuscript review, hence I rest this criticism by advising the authors to limit self-citation if possible.

My previous comments on handwaving arguments were not directed to the concepts of "static skin effect" and "Peierls distortion" themselves, but at how the authors use them merely as potential qualitative descriptions of their observations, lacking a quantitative basis and/or further proof. Including lines such as "Further theoretical investigations are required to shed light on this subject" and "Future studies will tell if this analogy plays any role in explaining our observation" do not remove the authors' responsibility to provide a solid scientific explanation/interpretation of their observations. Again polarising this statement for clarity: do the authors think that it is likely that their explanations given (not the experimental results) are completely wrong? Can detailed descriptions be given about the proposed theoretical work that needs to be carried out to support or falsify the provided interpretations, which the community can then specifically address in independent work? "Re-examine the idea of static skin effect" and "consider the contrast between the parity of the Bloch waves" are too broad comments to serve such purpose.

Finally, I acknowledge the authors' confidence in their experimental reproducibility, but any independent researchers will require a few additional details before being able to conduct the same experiments themselves. As the authors describe that the resistance measurements were carried out both with a home-made as well as a commercial set-up, can further details be given for both about the used current strengths as well as if they were based on a DC or AC technique (with the used signal frequency for the latter)? Moreover, can the brand of silver paste be added as well?

Further minor comments:

Might the 'nematicity'-related reference [51] (Science 354, 316, 2016) that was described in the Rebuttal Letter be better introduced in the relevant literature range in the Introduction?

Please re-read the sentence "It is known that *in* periodically driven Hamiltonian can provide topological protection in the so-called Floquet systems"

Reviewer #2 (Remarks to the Author):

The present version of the manuscript addresses several concerns raised by reviewers. The authors acknowledge that their manner of using the word "topology" in the previous title was somewhat misleading. In my opinion, the terms "protected by band symmetry" in the new title are still not experimentally proved, but given the significance of their experimental results, it is a minor point. In particular, it is surprising that the sample surfaces are not even polished. This observation indicates that the surface-dependent effect is robust. It would be interesting to investigate if the bulk properties, such as specific heat, are also surface-dependent. Therefore, I recommend this manuscript

for publication in Nature Communications.

What I would like to ask the authors is to carefully re-read the manuscript before submission. Even in modified sentences, I found typos like "higher then." I understand that the paper is very long, but this fact is no good excuse.

Reviewer #1 (Remarks to the Author):

As expressed in the previous reviewing round, I think that the authors' scientific work is worth publishing and reading and has several implications for existing research questions and a high likelihood to lead to follow-up research. Nevertheless, I still struggle with both the current presentation and interpretation of the work, to which the authors have only made very few changes with respect to their original manuscript.

Most statements made throughout the manuscript are scientifically supported by the authors' previous publications, not by those of others. As I highlighted previously, and now rephrased in a very polarised fashion for emphasis purposes: either the presented work is relevant to a large scientific audience and the fundamental academic work that it builds upon is cited at crucial stages throughout the manuscript, or this work merely builds on the authors' previous findings, which, although well-read, are only of interest to a small sub-set of this journal's readers. I am fully aware that eventually my previous statements only lead to a discussion based on subjective, research-ethical arguments, that might go beyond the scope of a manuscript review, hence I rest this criticism by advising the authors to limit self-citation if possible.

In the new version, we have included six new citations from other groups (ref. 13,14,15,16, 26 and 27 and deleted two references [Z. Zhu, B. Fauqué, Y. Fuseya, and K. Behnia, Phys. Rev. B 84, 115137 (2011) & Z. Zhu, B. Fauqué, K. Behnia, and Y. Fuseya, Journal of Physics: Condensed Matter 30, 313001 (2018)]. Suppressing other references to our previous works would damage our narrative. As the reviewer has noticed, our group has been one of the few working on transport properties of bulk bismuth in recent years. Let us recall what the Committee on Publication Ethics (COPE), a publisher-advisory body in London, (which has highlighted extreme self-citation as one of the main forms of citation manipulation) says about this:

While authors who self-cite their own work may be attempting to increase their citation rate artificially, there are good scholarly reasons for citing one's own previous work. The current manuscript which is under review may be on a continuum of a long-term programme of research and previous publications are relevant to understanding the history of the accumulated record. Further, not citing relevant previous work may result in allegations of self-plagiarism or redundant publications.

https://publicationethics.org/files/COPE_DD_A4_Citation_Manipulation_Jul19_SCREEN_AW2.pdf

My previous comments on handwaving arguments were not directed to the concepts of "static skin effect" and "Peierls distortion" themselves, but at how the authors use them merely as potential qualitative descriptions of their observations, lacking a quantitative basis and/or further proof. Including lines such as "Further theoretical investigations are required to shed light on this subject" and "Future studies will tell if this analogy plays any role in explaining our observation" do not remove the authors' responsibility to provide a solid scientific explanation/interpretation of their observations. Again polarising this statement for clarity: do the authors think that it is likely that their explanations given (not the experimental results) are completely wrong? Can detailed descriptions be given about the proposed theoretical work that needs to be carried out to support or falsify the provided interpretations, which the community can then specifically address in independent work? "Re-examine the idea of static skin effect" and "consider the contrast between the parity of the Bloch waves" are too broad comments to serve such purpose.

Our case for publication in Nature Communications is to report on a robust and important experimental observation, despite the absence of a satisfactory explanation. In order to focus on the observation, we have shortened the title. While the correlation between presence/absence of boundary conduction and symmetry/anti symmetry of the bands suggests some version of “quantum protection”, it does not necessarily imply it. We have also changed the last sentence of the abstract and reformulated the main paragraph in the concluding remarks in order to be even more explicit:

‘A satisfactory explanation of our results is missing and remains a challenge for theory. While the ‘static skin effect’ explains the existence of boundary conductance in a macroscopic crystal, it fails to explain its absence in antimony as well as the robust behavior of the conducting channel in presence of strong magnetic fields. We note that the semi-classical ‘static skin effect’ has not yet been formulated in a quantum-mechanical frame and the known contrast between the parity of the Bloch waves in Bi and in Sb at the L-point have not been taken into account. We argued that the latter may affect reflection at the crystal boundaries. To the best of our knowledge, this has not been yet incorporated in theory.’

We make it clear that an explanation for the whole set of experimental observations is missing. We also point to known loopholes in the theory, which may explain why this is the case. If any of these sentences are not accurate, we will be happy to correct them. The “theoretical work that needs to be carried out to support or falsify” is clearly identified. It is a quantum-mechanical (and not semi-classical) version of Azbel’s theory.

Finally, I acknowledge the authors’ confidence in their experimental reproducibility, but any independent researchers will require a few additional details before being able to conduct the same experiments themselves. As the authors describe that the resistance measurements were carried out both with a home-made as well as a commercial set-up, can further details be given for both about the used current strengths as well as if they were based on a DC or AC technique (with the used signal frequency for the latter)? Moreover, can the brand of silver paste be added as well?

We have added this information in the new version.

Further minor comments:

Might the ‘nematicity’-related reference [51] (Science 354, 316, 2016) that was described in the Rebuttal Letter be better introduced in the relevant literature range in the Introduction?

Done.

Please re-read the sentence “It is known that *in* periodically driven Hamiltonian can provide topological protection in the so-called Floquet systems”

Thank you, corrected!

Reviewer #2 (Remarks to the Author):

The present version of the manuscript addresses several concerns raised by reviewers. The authors acknowledge that their manner of using the word "topology" in the previous title was somewhat misleading. In my opinion, the terms "protected by band symmetry" in the new title are still not experimentally proved, but given the significance of their experimental results, it is a minor point. In particular, it is surprising that the sample surfaces are not even polished. This observation indicates that the surface-dependent effect is robust. It would be interesting to investigate if the bulk properties, such as specific heat, are also surface-dependent. Therefore, I recommend this manuscript for publication in Nature Communications.

We have changed the title, which has become “Boundary conductance in macroscopic bismuth crystals”. We agree with the reviewer that the correlation between band symmetry and the presence of the effect does not necessarily imply ‘protection’. We wish to stress the experimental result more than any interpretation of it. Nevertheless, as the reviewer notices, it is a remarkable fact that the effect does not need any surface treatment.

What I would like to ask the authors is to carefully re-read the manuscript before submission. Even in modified sentences, I found typos like "higher then." I understand that the paper is very long, but this fact is no good excuse.

Thank you, we have tried our best to correct other typos.